# Individuals 45 Years and Older in Opioid Agonist Treatment: A Scoping Review

**DOI:** 10.3390/ijerph22030458

**Published:** 2025-03-20

**Authors:** Zhanna Gaulen, Linn-Heidi Lunde, Silvia Eiken Alpers, Siv-Elin Leirvaag Carlsen

**Affiliations:** Department of Addiction Medicine, Haukeland University Hospital, 5009 Bergen, Norway; zhanna.gaulen@helse-bergen.no (Z.G.); linn-heidi.lunde@helse-bergen.no (L.-H.L.); silvia.eiken.alpers@helse-bergen.no (S.E.A.)

**Keywords:** opioid agonist treatment, older adults, ageing: substance use disorder, psychosocial factors, opioid use disorder, stigma, social isolation

## Abstract

This scoping review explores the unique challenges and needs faced by ageing individuals, aged 45 and above, in opioid agonist treatment (OAT), noting the earlier onset of age-related impairments among this population. The literature search was conducted in PsycINFO, EMBASE, MEDLINE, CINAHL, and Web of Science. A total of 28 observational studies were included. Five topics were identified: health, treatment, substance use, demographic, and social aspects. Findings reveal that, while health and age-related concerns are often discussed in the literature, limited attention has been given to gender differences, social factors such as financial issues, and psychological factors, which are also critical aspects of the lives of ageing individuals undergoing OAT. This review emphasizes the importance of expanding research to address these gaps, ultimately aiming to improve their overall well-being.

## 1. Introduction

Life expectancy of individuals treated for opioid addiction is increasing, largely due to the use of medications such as methadone and buprenorphine, usually in combination with psychosocial interventions [1,2,3,4,5,6]. This trend is evident in several countries [6,7,8]. In the EU, almost 70% of patients in OAT are aged 40 years or older [9], in Australia. the average age increased from 38 years in 2011 to 45 years in 2023 [10], while in Norway, the average age of OAT patients increased from 39 years in 2004 to 48 years in 2022 [11], with projections suggesting a continued rise.

However, this ageing population faces a range of health and social challenges for various reasons. Firstly, the lasting impact of past substance use may contribute to an increased vulnerability to chronic diseases, even after cessation. Conditions such as cardiovascular disease, liver dysfunction, and respiratory problems are prevalent [12]. Furthermore, the natural ageing process increases the risk of cognitive impairment and dementia, which can complicate treatment adherence and hinder recovery [13,14]. Pre-existing health conditions and polypharmacy—the concurrent use of multiple medications—further increase the risk of adverse drug interactions and complicate care for these patients [15].

Moreover, social and emotional factors affect the well-being of ageing OAT patients. While many individuals over the age of 45 in OAT appear to have a good financial situation, self-reported poverty persists among some in this age group, and many express dissatisfaction with their financial circumstances [16]. A Norwegian study of OAT patients found that financial satisfaction was the single most influential factor affecting quality of life over time [17]. Many also experience increased isolation and loneliness, often stemming from the loss of peers and partners as well as changes in social roles [18]. The persistent stigma associated with both addiction and ageing can further amplify their sense of isolation, limiting community engagement and negatively impacting mental health and self-identity [16,19,20,21]. Evidence indicates a discrepancy of approximately 15–20 years between biological and chronological age in this population, with life expectancy roughly 20 years shorter than that of non-dependent individuals [22,23].

Research on older adults in OAT remains limited [24,25]. The definition of “older adults with opioid use disorders (OUDs)” varies internationally, with no standard age cut-off. Typically, the threshold is set below 60 years [26]; European studies may use 40 years [26], while U.S. studies often use 50 years [23,27]. This review employs a threshold of 45 years, as individuals with OUDs frequently exhibit signs of ageing earlier than the general population [24,26]. The review aims to:Identify characteristics of OAT patients aged 45 and above;Explore key social, psychological, and medical factors influencing this population;Highlight gaps in current research regarding their needs and challenges.

This review lays the foundation by summarizing existing research for clinicians and practitioners in the field and offering recommendations for future research and healthcare interventions to enhance the quality of care and outcomes for this growing demographic.

## 2. Materials and Methods

A research team with expertise in OUD, geropsychology, and qualitative and quantitative methodologies was assembled. A scoping review methodology was chosen to explore the breadth of evidence on this topic [28] as it effectively summarizes research, clarifies concepts, assesses methodologies, identifies relevant characteristics, and highlights knowledge gaps to guide future research [25,28].

To ensure methodological rigor, the team followed the Joanna Briggs Institute (JBI) methodology and the PRISMA-ScR (Preferred Reporting Items for Systematic Reviews and Meta-Analyses extension for Scoping Reviews) [29,30]. A study protocol is registered with the Open Science Framework (OSF) under the project identifier osf.io/p3raz.

### 2.1. Eligibility Criteria

The inclusion and exclusion criteria are outlined in Table 1.

The age of adults in OAT was the primary criterium for inclusion. Studies were included if they focused on age and age-related outcomes in their objectives or findings. In studies with a mean participant age of 45 that also included individuals younger than 45, only findings explicitly concerning participants aged 45 or older were included. Data from articles that specifically referenced this age group were incorporated into our scoping review.

Conference summaries, abstracts, and posters were excluded to maintain rigor and reliability in the review process. Reviews were omitted to avoid duplication and to focus on primary research articles presenting original data. Grey literature (e.g., reports, theses) was not included due to variability in quality and lack of peer review, despite its potential to offer valuable insights. Meta-analyses were excluded to prevent overlap with journal publications. Case studies were excluded because their focus on individual cases or small samples did not align with the review’s objective to synthesize broader evidence. Interventional studies, particularly randomized controlled trials (RCTs), were also excluded because they focus on evaluations of intervention efficacy. Only studies in English, German, and Scandinavian languages were included, a decision based on the research team’s language proficiency. This approach was chosen to mitigate limitations in translation resources and ensure thorough examination and synthesis of the literature.

### 2.2. Information Sources and Search Strategy

The literature search was conducted across PsycINFO, EMBASE, MEDLINE, CINAHL, and Web of Science. The primary search was carried out on 17 January 2022, with supplementary searches conducted on 9 January 2023, 25 September 2023, and 26 September 2024. No date limits were applied. Key terms were used to capture studies related to three areas of interest: age, opioid treatment, and types of substances. These terms were connected using the Boolean operators (OR/AND). The search terms were adapted for the databases according to their specific parameters (Table 2). Additionally, the reference lists of included articles were screened to identify further relevant studies.

### 2.3. Study Selection

Titles were screened using EndNote version 20 (Clarivate Analytics, Philadelphia, PA, USA) by the lead investigator to remove duplicates. The remaining titles were exported to Rayyan [31] for further screening. Four investigators independently reviewed abstracts for inclusion, with disagreements resolved through discussion. The team collaborated closely throughout the screening process to ensure a consistent approach. To strengthen the robustness of the selection process, each included article was reviewed by at least two team members. Full-text reviews were conducted on the remaining articles to determine eligibility based on inclusion criteria. The results of the selection process are presented in a PRISMA-ScR flow diagram (Figure 1).

### 2.4. Data Extraction and Analysis

Data extraction was performed by four investigators using an electronic Excel form covering title, author, publication year, demographics, methodology, results, and limitations.

Inductive content analysis was applied to identify common themes across the included studies, with the primary goal of categorizing and exploring recurring themes and patterns. These themes were then analysed in relation to the objectives of the scoping review. In line with the scoping review methodology [30], no formal quality assessment of the included studies was performed, as this was not required to meet the specific objectives of this review. Findings are presented in a descriptive summary, supported by tables and figures where appropriate.

## 3. Results

### 3.1. Search Results

Database searches identified a total of 11,105 titles and abstracts. After removing duplicates, a total of 7801 articles remained for screening. Of these, 159 articles were identified as potentially eligible for inclusion. Following a full-text review, 25 studies met the inclusion criteria. An additional three studies were identified through a reference list review of the included studies, bringing the total number of studies in this review to 28 (Figure 1).

### 3.2. Study Characteristics

Articles published up to 26th September 2024 were included. The first identified publication focusing on older adults in OAT was recorded in 2005. The number of publications shows a fluctuating pattern that highlights an inconsistent yet recurrent interest in the topic. Most studies, 15 of 28, employed a quantitative cross-sectional design and were conducted in high-income countries such as the United States and the UK (Table 3).

Of the 28 studies included, 16 focused on participants aged 50 and above [23,32,33,34,35,36,37,38,39,40,41,42,43,44,45,46]. Of these, two longitudinal studies tracked participants over 25 years, from baseline interviews conducted between 1978 and 1981 to follow-ups from 2005 to 2009. Twelve studies included participants aged 18 or older, with a mean age of 45 [47,48,49,50,51,52,53,54,55,56,57,58].

Age thresholds and justifications for focusing on older adults in OAT varied across studies (see Table 4), but several common themes emerged (Table 5). These included the growing proportion of older adults within OAT, a rise in age-related health risks, and comorbidities such as chronic illnesses, social isolation, and healthcare disparities. Several studies cited the ageing “Baby Boomer” cohort [23,39,41,44,46,47,51,59], whose higher rates of substance use necessitate tailored services to address specific health and social needs.

Even though research aims differed, 15 studies focused on examining age-related health outcomes, such as mortality rates and chronic conditions. Five studies assessed healthcare utilization and proposed considered potential modifications to current OAT practices to better serve an ageing population (see Table 5).

The studies involved several key comparison groups. Common comparisons included older adults (>50) versus younger adults (<50). Other comparisons included OAT patients versus the general population to evaluate chronic health conditions and mortality rates. Gender differences were also explored, focusing on health outcomes and treatment adherence among men and women. Additional comparisons involved different substance use, specific health conditions (chronic obstructive pulmonary disease, erectile dysfunction), and different age groups (Table 4).

In these 28 studies, five core themes emerged: somatic and mental health (*n* = 15; 53%), treatment characteristics (*n* = 5; 18%), demographic factors (*n* = 3; 11%), social aspects (*n* = 3; 11%), and substance use patterns (*n* = 2; 7%). These topics are presented in Table 5 with each row representing one study and its key results (see Table 5).

Some articles had several subthemes (e.g., health, substance use, and demographics); however, they were categorised after their main focus (Figure 2).



*Health*



Health was the predominant focus, with studies addressing chronic conditions commonly seen in this demographic, such as COPD [23,38,39,40,46,48,50], hepatitis C [34,52,55], hypertension, and pain-related problems [35]. A significant finding was that mortality in this age group often results from somatic illnesses rather than overdose [60], with deaths often occurring at home [47].



*Treatment*



Studies on treatment characteristics examined both the accessibility and effectiveness of OAT in older patients. Barriers to treatment included physical mobility issues, transportation limitations, and age-related stigma within OAT programmes, all of which can impact retention rates and adherence. Recommendations for more age-adapted OAT services were a common theme in these studies.



*Demographic factors*



Most of the studies included both men and women, indicating a gender-inclusive approach. However, only two studies reported a higher proportion of female participants [37,42]. Most participants were unemployed and reliant on social welfare, though they generally had stable housing [33].



*Social aspects*



Studies examining social aspects focused on stigma, loneliness, and the effects of bereavement and social isolation on this ageing population. Social factors were identified as significant barriers to treatment adherence and well-being [34,37,42,44,57], with some studies indicating that a lack of social support could exacerbate psychological distress and isolation [33,34,38,41,46].



*Substance use*



While illicit drug use appears to decline with age among OAT patients (either becoming non-existent, or occurring less frequently), studies reported an increase in alcohol [34,35,39,41,51,53,57] and benzodiazepine consumption [33,39].

## 4. Discussion

This scoping review aimed to examine the characteristics and factors influencing older adults, aged 45 and above, in OAT and to identify research gaps to improve care and outcomes for this population. Despite identifying over 158 potential articles, only 28 were included in the final analysis, indicating the limited focus on older adults in OAT. Nonetheless, the recent rise in publications suggests a growing recognition of this population’s specific needs, reflecting the global trend of an ageing OAT demographic. This review focused on five main categories: health, social aspects, demographics, treatment, and substance use.

### 4.1. Health

The articles included in this review primarily address various health-related issues [23,34,35,38,39,40,41,43,46,47,48,50,52,55,56], which is not surprising given the emphasis on the older population. However, compared to the extensive body of knowledge on ageing and diseases commonly observed in the general older population, there is much less understanding of how these health challenges manifest in the older OAT population. For instance, methadone metabolism and renal clearance decline with age, potentially increasing the risk of overdose and other adverse outcomes [61,62]. Furthermore, older patients are generally more likely to be prescribed multiple medications for comorbid conditions, heightening the risk of drug–drug interactions [63]. Such interactions may contribute to the elevated rates of drug-related deaths in this demographic [64].

### 4.2. Treatment

Furthermore, the organisation of OAT services also plays a pivotal role in promoting health equity. Integrating OAT with broader healthcare services improves older OAT patients’ access to treatment for co-occurring somatic and mental health conditions [58], such as hepatitis C [65]. Despite this, only 25% of adults with OUD and mental health disorders received treatment for both in the past year [66]. We found that evidence supports integrating psychosocial therapy with medications for opioid addiction as a means to enhance clinical outcomes [67].

Moreover, research highlights that healthcare professionals’ education level, experience, and knowledge of OAT play a critical role in the stigma OAT patients encounter [68,69]. Addressing stigma and promoting health equity are therefore essential for developing effective interventions. Establishing a supportive, non-judgmental environment is vital not only for encouraging older adults to seek treatment but also for fostering resilience and recovery within this vulnerable population.

### 4.3. Demographics

We also found that gender-specific challenges are another underexplored area in OAT research. Women and men may have significant differences in their treatment experiences, facing distinct challenges in attaining long-term recovery [70]. While men are often overrepresented in substance use studies, there is a pressing need to explore additional gender-specific issues beyond erectile dysfunction among older adults [55,56]. Furthermore, our findings reveal a lack of studies focusing exclusively on older women in OAT. Women in this population face unique challenges, including menopause-related symptoms and, for some individuals, issues associated with a history of sex work.

In this scoping review, considerable difficulty was encountered in establishing an age cut-off for defining “older adults” in OAT, as studies used varying age thresholds. Some studies in this review also involved younger OAT patients [23,47,48,49,50,51,52,53,54,55,56,57,58], but only results explicitly focused on participants aged 45 or older are presented, ensuring the applicability of the findings to our target demographic. The inconsistency in age definitions complicates cross-study comparisons and limits the ability of healthcare systems to provide targeted care. Previous systematic reviews [6,25] also report wide-ranging definitions of “older adults” among people who use drugs, ranging from 40 to 70 years. These variations may reflect different cultural interpretations of ageing, as societal values shape ageing experiences differently across populations [71]. Furthermore, any chosen cut-off point is inherently subject to debate: some stakeholders argue for a lower threshold (e.g., 40–45 years) to capture early ageing effects in people who use drugs [25,26], while others advocate for an age closer to traditional retirement milestones [72]. Developing unified frameworks and clear, consistent definitions is essential to better characterise this population and enable effective comparisons.

### 4.4. Social Aspects

We found that OAT research remains heavily biased toward health and medical issues, while social and psychological factors, such as isolation, stigma, housing instability, financial difficulties, and transportation difficulties, are underexplored [41].

Being both poor and older was perceived as particularly stigmatising [37]. The financial structure of OAT plays a crucial role in patients’ economic circumstances. While some OAT clinics in certain countries, such as those in Scandinavia, are publicly funded—either fully or partially—others, including those in the UK, Canada, the US, and Australia, operate within the private sector. This may involve private clinics or GPs prescribing medication dispensed at community pharmacies [73]. An Australian study, with a mean participant age of 41 years (SD = 8.7), found that out-of-pocket expenses, such as travel costs and OAT-related fees, can act as barriers to treatment [64]. Although, none of the included studies specifically examined the financial circumstances of older adults in OAT, our findings indicate that they face economic challenges that restrict access to essential resources and hinder their ability to advocate for and receive equitable care [37]. Individuals with low incomes in the general population are consistently more affected by loneliness than those in higher socioeconomic groups [74]. It is likely that similar dynamics will apply to older individuals in OAT as well. This scoping review highlights the lack of attention given to financial aspects in this population, underscoring the importance of addressing economic factors and their potential connections to mental health, healthcare access, and overall quality of life.

Misconceptions and stereotypes portraying individuals with substance use disorders as aggressive, problematic, or neglectful contribute to stigma. We found that such attitudes perpetuate an “us-versus-them” mentality, reinforcing social distance and deepening biases for older OAT patients. Our results also showed that older individuals in OAT report being treated as second-class citizens due to receiving different—and often inferior—care [37,44]. For instance, they may be offered dentures instead of dental implants, as dentures are a more cost-effective option [75]. The perception of being regarded as second-class citizens can be further reinforced by the self-isolation of older OAT participants [42].

Stigma serves as a barrier to accessing healthcare and seeking treatment [76]. Individuals in OAT, regardless of age, often experience stigmatization from both society and healthcare professionals, including those working in pharmacies, drug treatment clinics, dental practices, and hospitals [32,37,69]. We found that this stigma complicates care and worsens outcomes for older adults undergoing OAT [37]. Furthermore, our results showed that negative perceptions associated with substance use disorders often lead to feelings of shame, social isolation, and a lack of support from both the community and healthcare providers, further exacerbating health disparities [77].

As a result of our findings, older adults undergoing OAT encounter barriers tied to generational norms emphasizing self-reliance in managing health. A lack of awareness about the specific needs of older OAT patients, combined with their reluctance to disclose their treatment status due to fear of stigma or discrimination, exacerbates existing health inequities [37]. These interrelated factors significantly undermine health equity for this population.

To understand the trajectory of ageing in OAT and to inform interventions that adapt to changing circumstances longitudinal research is necessary. Our review revealed two studies [45,46] that examined the long-term outcomes of OAT in older adults, particularly in terms of quality of life. A person’s quality of life is significantly influenced by their social connections with family and friends [78]. Additionally, engaging in hobbies or meaningful activities can provide a sense of purpose and help alleviate the loneliness commonly reported by many older OAT patients [17]. In addition, as stable housing may become more relevant and attainable for older individuals in OAT, it may significantly impact their quality of life, health, and social belonging [79,80].

### 4.5. Substance Use

Substance use patterns among OAT patients evolve with age. While illicit drug use in older OAT patients often declines [20], alcohol and benzodiazepine consumption may increase, likely influenced by reduced social networks and heightened feelings of loneliness [39]. Our results indicate that older individuals in OAT hold a particular role and status within substance-using communities, making it challenging to build new identities and establish new social networks outside these environments, especially as they age [42]. Using substances to numb emotions is a well-known coping strategy among people with SUDs [81,82,83], and for some, using alcohol to dull the pain of loneliness or to cope with trauma from a long history of substance use may feel like a viable option.

### 4.6. Methodological Aspects and Future Research Directions

The studies included in this scoping review vary significantly in methodological rigor, representativeness, and overall suitability for investigating the unique needs of older adults in OAT. Sixteen studies employed observational and cross-sectional designs, utilizing healthcare registries and national datasets that provide descriptive insights into broad trends in health outcomes and treatment trajectories. Conversely, longitudinal studies, though only two in number, offer evidence on the progression of comorbidities and the long-term effects of OAT.

However, the included studies are limited in several key respects. Cross-sectional designs cannot establish causality or track changes over time, a significant limitation when studying trajectories of older adults. As a result, the long-term effects of OAT, particularly concerning quality of life and healthcare utilization, remain poorly understood. Similarly, many smaller qualitative and clinic-based studies rely on convenience sampling, which introduces potential biases as participants may not fully represent the population of older adults. 

From a methodological perspective, the included studies demonstrate uneven suitability for addressing the needs of older adults in OAT. Large registry-based studies offer strong generalizability but often lack the depth required to capture individual experiences and the contextual factors that influence outcomes. Conversely, qualitative studies provide valuable insights into contextual factors but are frequently limited by small sample sizes and localized recruitment. The absence of mixed-methods approaches, which could integrate the strengths of both quantitative and qualitative methods, further reduces the ability of this body of work to fully capture the multifaceted nature of ageing in OAT. 

Nevertheless, the included studies provide important insights while underscoring methodological limitations, emphasizing the need for more rigorous and inclusive research. Future studies should adopt longitudinal designs, intersectional analyses, and mixed-methods approaches to better address the complex needs of older adults in OAT. Bridging gaps between health, social, and systemic factors could more effectively inform policies and practices that promote health equity and improve outcomes for this vulnerable population. 

Collaboration among healthcare providers, researchers, and policymakers is crucial to meeting the needs of an ageing population. Holistic, age-appropriate OAT that integrates chronic disease management, flexible treatment options, mobility support, mental health care, and social services can reduce isolation, mitigating stigma, and enhancing adherence, ultimately improving patient outcomes.

### 4.7. Strengths and Limitations

Our work is distinct in its exclusive focus on OAT patients. This review provides valuable insights, especially for healthcare providers and policymakers aiming to improve care for older adults in OAT.

The exclusion of grey literature and interventional studies limits the breadth of the findings but ensures a focus on peer-reviewed studies that provide robust, contextually detailed insights. Although grey literature may offer diverse perspectives, its inconsistency across political and social settings makes standardization difficult. RCTs, which typically focus on treatment and medication, provide insights into clinical effects rather than the broader social, psychological, and physical changes associated with ageing. By not including reviews, we may have overlooked high-quality analyses that could have strengthened our findings. Likewise, excluding non-translated works may have unintentionally narrowed the evidence base, particularly in a global research landscape where significant findings are often published in multiple languages. By relying on peer-reviewed studies, this review strengthens the reliability of its conclusions within established research parameters.

Furthermore, the small sample sizes in many studies made it challenging to draw broad conclusions about this population, limiting the generalizability of the findings. While the quality of the included studies was not formally assessed, scoping reviews aim to provide a broad overview of the research landscape, capturing underexplored areas rather than offering in-depth evaluations [84].

Most included studies were conducted in high-income countries, which limits the generalizability of the findings. However, this concentration provides valuable insights into the complexities of ageing in OAT within well-established treatment frameworks, setting a benchmark for future studies in diverse socioeconomic contexts.

## 5. Conclusions

Research on older adults in OAT has increased over the past decade, though it remains heavily focused on the health and disease aspects of ageing. There is limited knowledge of how these individuals experience ageing and how it affects their quality of life and rehabilitation. To address these gaps, more comprehensive, longitudinal studies are needed to capture the complex experiences of this population and guide the development of tailored interventions. This review highlights the unique challenges faced by older adults in OAT, underscoring the need for specialized healthcare services that address their physical, psychological, and social needs. As the population of older adults with opioid use disorder continues to rise and age, we need further research on all aspects of their lives in OAT.

## Figures and Tables

**Figure 1 ijerph-22-00458-f001:**
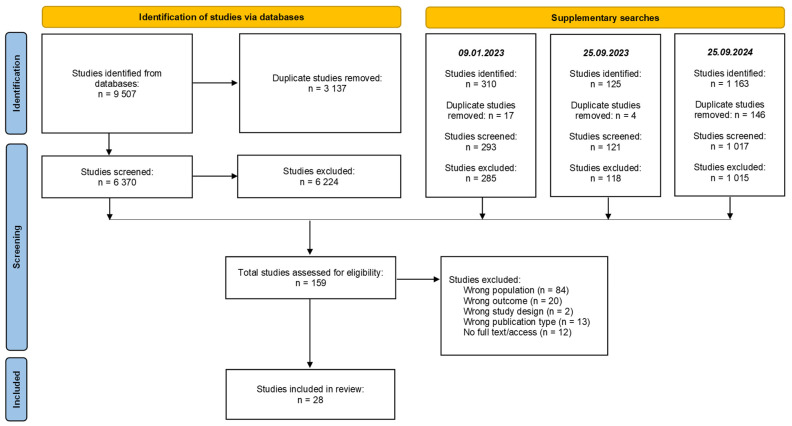
PRISMA-ScR flow diagram of the identification, screening, and inclusion of studies.

**Figure 2 ijerph-22-00458-f002:**
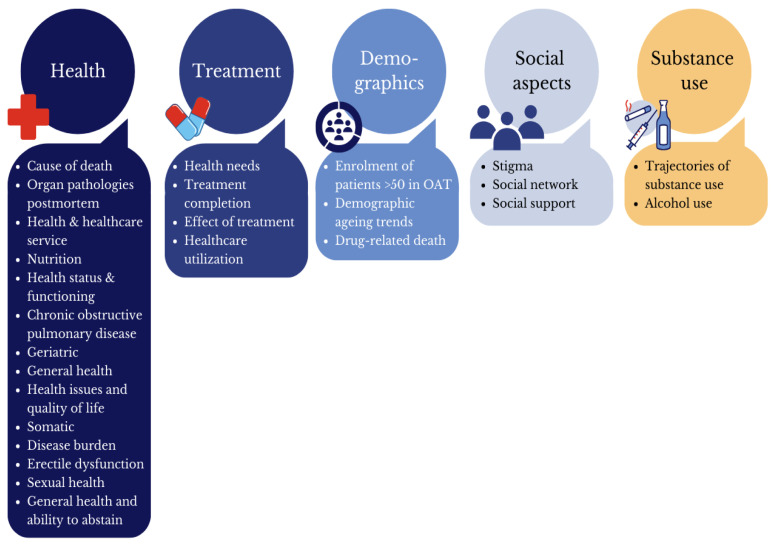
Overview of main and subtopics.

**Table 1 ijerph-22-00458-t001:** Inclusion and exclusion criteria for scoping review on older adults with OUD in OAT.

Parameter	Inclusion Criteria	Exclusion Criteria
** *Population* **	Individuals (aged 45 and above) undergoing treatment for OUD	Individuals in OAT under the age of 45
** *Concept* **	Studies focusing on all aspects of life	Studies focusing solely on younger adults, other treatments (e.g., cancer therapy), or not related specifically to ageing in OAT
** *Context* **	Inpatient or outpatient settings, including primary care, addiction treatment centres, or healthcare settings in any country	Studies on individuals not in OAT
** *Study Design* **	Observational studies (prospective/retrospective) and qualitative studies (interviews, focus groups)	Abstracts, posters, annual meetings, editorial commentary, reviews, reports, grey literature, meta-analyses, case studies, interventional studies
** *Time Frame* **	No time restrictions were imposed	
** *Language* **	English, Norwegian, Swedish, Danish, or German	

**Table 2 ijerph-22-00458-t002:** Search strategy for the scoping review in the EMBASE research database.

** *EMBASE search sting* **	aged/OR middle aged, (aged or elder* OR aging OR ageing OR senior OR “older patient*”), opiate/OR methadone, buprenorphine/OR diamorphine, AND (maintenance OR replacement OR replacing OR substitution OR substitute* OR assisted OR medicated), ((opioid OR opioide OR opiate* OR methadone OR metadon OR buprenorfine OR buprenorphine OR buprenorphin OR diamorphine OR heroin OR heroine OR metadone) adj3 (maintenance OR replacement OR replacing OR substitution OR substitute* OR assisted OR medicated)

**Table 3 ijerph-22-00458-t003:** Studies on adults in opioid treatment by country, methodology, and data collection methods, with some articles employing multiple methods of data collection.

Study Characteristics	N (%)
Country	
U.S.	13 (46.4)
UK	4 (14.2)
Norway	3 (10.7)
Malaysia	2 (7.1)
Australia	1 (3.6)
Canada	1 (3.6)
China	1 (3.6)
Spain	1 (3.6)
Sweden	1 (3.6)
Switzerland	1 (3.6)

**Table 4 ijerph-22-00458-t004:** Studies on older adults in opioid treatment: methodological approaches, justifications for age, and research aims.

Author ^1^, Year, Country	Methodology, N, Sex, Age (Range, SD, Mean)	Rationale for Focusing on Older Adults and Age	Aims
**Ayres et al., 2012,** **UK [32]**	Qualitative, semi-structured interviews, focus group; descriptive design. N = 20; 85% men. Age range: 55–66.	(a) Increasing trend of older adults in drug treatment services; (b) Neglected and overlooked population.	1. Explore health needs and treatment experiences; 2. Ideas for developing service.
**Badrakalimuthu** **et al., 2012,** **UK [33]**	Quantitative, retrospective cross-sectional design. Group comparison: <50 vs. >50. N = 92; 83% men. Age range: 50–73.	(a) To ensure adequate sample size for the study; (b) Similar age range used in previous studies.	1. Explore characteristics of patients >50 entering treatment; 2. Compare these characteristics with those <50.
**Bech et al., 2019,** **Norway [47]**	Quantitative, national, observational study; registry data. N = 200; 74% men. Mean age: 48.9 (SD = 8.4, range 23–71).	(a) Norway has one of Europe’s oldest OAT populations; (b) Somatic causes of death are likely to rise as OAT patients age.	1. Describe causes of death; 2. Estimate all-cause and specific crude mortality rates by age, OAT medication, and gender; 3. Explore characteristics associated with drug-induced death vs. other causes.
**Bech et al., 2021,** **Norway [48]**	Quantitative, national, cross-sectional design. N = 122; 75% men. Mean age: 48 (SD = 8.7, range 23–68).	(a) OAT patients are ageing; (b) Few post-mortem data on organ pathology in OAT patients over 40; (c) Few data on prevalence of enlarged organs.	1. Document organ pathologies post-mortem; 2. Estimate the relationship between individual characteristics and pulmonary, cardiovascular, hepatic, or renal pathologies.
**Beynon et al., 2009,** **UK [34]**	Qualitative, semi-structured interviews; descriptive design. N = 10; 90% men. Age range: 54–61.	(a) Public health concern about older drug users; (b) Unique health challenges; (c) Underrepresentation in research.	1. Identify substances used; 2. Assess self-reported health; 3. Examine contact with generic and specialist health services.
**Beynon et al., 2013,** **UK [35]**	Qualitative, semi-structured interviews; pilot study. N = 10; 90% men. Age range: 51–61.	(a) Age-related health changes require more attention; (b) Need to address nutritional deficiencies to improve health outcomes.	1. Investigate diet perceptions; 2. Investigate influences on food choices; 3. Identify if body mass is considered in care planning.
**Choi et al., 2022,** **USA [36]**	Quantitative, national, cross-sectional design, retrospective cohort element. Group comparison: (a) primary heroin vs. prescribed opioids; (b) 55–64 vs. 65+. N = 127,034; 72% men. Age: >55.	(a) Rising cases of OUD among older adults from 2004–2015; (b) High prescription opioid use in older adults due to chronic pain; (c) Health risks associated with opioid use in older adults.	1. Examine treatment completion rates by setting (detoxification, residential, and outpatient); 2. Identify factors associated with treatment completion; 3. Analyse substance use patterns; 4. Evaluate treatment characteristics.
**Conner et al., 2008,** **USA [37]**	Qualitative, semi-structured interviews; descriptive design. N = 24; 42% men. Age range: 50–60.	(a) Underrepresentation of older adults in stigma research; (b) Experiences of ageing methadone clients; (c) Healthcare disparities.	1. Examine stigma experiences; 2. How multiple stigmas delay entry to substance abuse and mental health treatment.
**Fareed et al., 2009,** **USA [49]**	Quantitative, retrospective cohort study. N = 91; 98–100% men. Age 40+. Mean age: 57 (SD = 3) deceased; 57 (SD = 9) retained; 53 (SD = 6) dropped out.	(a) Experiences of the effects of ageing and opioid dependence; (b) Comorbidities compound the health risks associated with opioid dependence; (c) Premature mortality differs between older adults and younger patients.	1. Identify key factors contributing to premature death; 2. Assess MMT effectiveness in reducing drug use and improving overall health outcomes; 3. How comorbid conditions and medical issues impact health and mortality.
**Grella & Lovinger, 2011, USA [45]**	Quantitative, longitudinal retrospective cohort design. Group comparison: Men vs. women. N = 343; 55.7% men. Mean age: 58.3 (SD = 4.9) men; 55.0 (SD = 4.1) women.	(a) Long-term outcomes and changes in drug use patterns; (b) Impact of childhood factors on heroin addiction; (c) Explore substance use disorders and ageing.	Examine long-term drug use trajectories in MMT.
**Grella & Lovinger, 2012, USA [46]**	Quantitative, longitudinal retrospective cohort design. Group comparison: (a) Men vs. women; (b) MMT vs. general population, N = 343; 55.7% men, Mean age: 58.3 (SD = 4.9) men; 55.0 (SD = 4.1) women.	(a) “Baby Boomer” health problems and service needs; (b) Rising substance use problem in adults 50+.	Assess overall health status and psychosocial functioning in older adults with a history of heroin dependence.
**Grischott et al., 2019,** **Switzerland [50]**	Quantitative, cross-sectional design. Group comparison: Older adults with chronic obstructive pulmonary disease (COPD) vs. younger adults without COPD. N = 125; 76% men. Mean age: 45.1 (SD = 9.3, range 22–65).	(a) Explore chronic illnesses in ageing OAT patients; (b) Older adults face age-related health issues.	1. Estimate COPD prevalence and risk factors; 2. Establish the distribution of airflow obstruction severity; 3. Compare COPD rates in different age groups with the general population; 4. Assess willingness to adopt lifestyle changes and use therapies among OAT patients with COPD.
**Han et al., 2015,** **USA [51]**	Quantitative, descriptive retrospective cohort design. N = 37,038 (1996); 40,328 (2003); 34,270 (2012); 66–69% men for all ages. Five age groups (≤40, 41–49, 50–59, 60–69, and ≥70).	(a) Variability in age cut-offs in previous studies; (b) Studies using large national datasets generally defined older adults as those aged 50 or 55 old.	1. Examine age trends in older adults in OAT programmes in New York; 2. Characterize demographics, substance use, and physical impairments.
**Han et al., 2022,** **USA [38]**	Quantitative, descriptive cross-sectional design. Group comparison: MMT vs. national population. N = 47; 76.6% men. Mean age: 58.8 (SD = 5.8, range 50–75).	(a) MMT individuals ≥50 are more likely to have age-related health conditions and impairments; (b) Older adults have unique healthcare needs compared to younger people; (c) The age range involves prolonged opioid use, increasing vulnerability to geriatric conditions.	Determine prevalence of geriatric conditions in older MMT adults.
**Han et al., 2024,** **USA [44]**	Qualitative, descriptive phenomenological design. N = 36 OUD; 69.4% men. Mean age: 63.4 (SD = 5.1, range 55–77).	(a) A national initiative to enhance care for older adults by creating age-friendly health systems grounded in evidence-based geriatric principles; (b) a first step to developing age-friendly OUD models of care.	1. To explore the ageing experience with OUD; 2. Barriers to medical care for older adults who receive care in OTPs.
**Lintzeris et al., 2016,** **Australia [39]**	Quantitative, cross-sectional design. Group comparison: Older vs. younger adults. N = 99 in the older group (n = 69 in OAT, n = 30 in alcohol treatment); 74% men. Mean age: 55 (SD = 4.5, range 50–71).	(a) More older patients in drug and alcohol treatment in Australia; (b) Projected rise in adults 50+ due to longer life expectancy and higher substance use rates; (c) Address health and social consequences of substance use in older adults.	1. Examine substance use, health, cognition, social function, and health service use in older clients (≥50 years); 2. Compare these measures with younger clients.
**Lofwall et al., 2005,** **USA [23]**	Quantitative, cross-sectional design. Group comparison: 50–66 vs. 25–34 years old. N = 41 older participants (total n = 67); 51% men in the older group, 58% in the younger group. Mean age: 53.9 (SD = 0.6, range 50–66).	(a) The “Baby boomer” generation has higher substance use rates; (b) Substance use problems in older adults are an “invisible epidemic”; (c) Rising need for specific treatment for older adults; (d) Lack of understanding of non-alcohol drug use in older adults; (e) Identify specific challenges and treatment needs for older adults in substance abuse services.	1. Compare older vs. younger opioid-dependent patients on psychiatric, substance use, medical, legal, and psychosocial variables; 2. Compare their health status to age- and sex-matched U.S. norms.
**Maruyama et al., 2013,** **Canada [40]**	Quantitative, case-control design. Group comparison: MMT patients vs. non-MMT patients. N = 199 MMT patients (total n = 398); 71.9% men. Mean age: 59.4 (SD = not given; 50+).	(a) Limited research on older MMT patients’ health compared to the general population; (b) Tailored healthcare services needed for older patients; (c) Care coordination issues due to the separation between family and methadone doctors; (d) Older patients face barriers to accessing primary healthcare and social services.	Compare prescription rates for hypertension, COPD, diabetes, and depression between MMT patients vs. matched controls.
**Medved et al., 2020,** **Norway [52]**	Quantitative, cross-sectional design. N = 156; 59.6% men. Mean age: 47.9 (SD = 7.1, range 31–64).	(a) Increasing age of OAT patients in Norway; (b) Chronic somatic health conditions increase with ageing; (c) Lack of data on long-term OAT patients’ somatic health needs.	1. Identify chronic conditions, health care use, and treatment satisfaction; 2. Explore self-reported somatic disease burden and its associated factors.
**Nyamathi et al., 2009,** **USA [53]**	Quantitative, cross-sectional design. Group comparison: (a) ≥50 years vs. <50 years; (b) MMT moderate vs. heavy alcohol users; (c) Fair/poor vs. good/excellent health status. N = 190; 57.5% men. Age range: 18–55 (62% 50+).	No rationale for focusing on older patients; but dichotomized age at 50 for analysis purposes.	1. Describe alcohol use prevalence among MMT patients; 2. Assess correlates of alcohol use, including demographics, health care use, social support, psychological resources, and risk factors.
**Pierce et al., 2018,** **UK [54]**	Quantitative, retrospective national cohort study. Group comparison: 18–24; 25–34; 35–44; 45–64 years old. N = 129,979; 77% men of the n = 1266 drug-related deaths (DRDs). Age range: 18–64 at inclusion.	(a) Methadone-specific DRDs increase with age; (b) Older opioid users more likely to have comorbid physical and mental health conditions; (c) Methadone’s pharmacodynamics vary with age; (d) Need for targeted monitoring and interventions for older methadone clients.	1. Analyse demographic risk factors affecting hazard ratio for DRDs, methadone-specific DRDs, and heroin-specific DRDs; 2. Adjust analysis for treatment periods and declared substance misuse; 3. Pool age-related hazard ratios for methadone-specific deaths from the Scotland and England cohorts.
**Ramli et al., 2019,** **Malaysia [55]**	Quantitative, questionnaire-based study, cross-sectional design. N = 50; 100% men. Age 18+. Mean age: 47.3 (SD = 7.5) in erectile dysfunction treatment, 51 (SD = 10.2) in no treatment.	No rationale for focusing on older patients but examined age associations with treatment-seeking behaviour.	1. Examine health-seeking behaviour for erectile dysfunction among MMT patients; 2. Assess factors and effectiveness of erectile dysfunction treatment-seeking behaviour.
**Ramli et al., 2020,** **Malaysia [56]**	Quantitative, cross-sectional design. N = 271; 100% men. Mean age: 48.8 (SD = 9.7, range 25–71).	(a) Sexual activity decreases with age in the general population; (b) MMT is linked to sexual dysfunction, but knowledge of sexual inactivity is limited.	Determine prevalence and risk factors for sexual inactivity.
**Rosen et al., 2008,** **USA [41]**	Quantitative, cross-sectional design. N = 140; 64.3% men. Mean age: 53.9 (SD = 40.1, range 50–67).	(a) Substance abuse among older adults is a growing public health issue; (b) Ageing opiate users face exacerbated health problems and social isolation; (c) Understanding comorbid conditions is crucial for serving an ageing population.	1. Document general medical and mental health disorders; 2. Evaluate the ability to remain abstinent later in life.
**Shen et al., 2018,** **China [57]**	Quantitative, cross-sectional design. N = 324; 76.9% men. Mean age: 45.2 (SD = 6.3; 20+).	Age emerged as a key variable in analysing MMT clients’ drug use behaviours.	1. Examine social network characteristics; 2. Explore relationships between social networks and drug use behaviours; 3. Identify protective and risk factors in social networks affecting drug use; 4. Provide insights into social networks and drug use.
**Smith et al., 2009,** **USA [42]**	Qualitative, descriptive design. N = 24; 41.7% men. Mean age: 58.4 (range 52–68).	(a) National data show significant adults over 50 receiving treatments for heroin use; (b) More older substance users are seeking help; (c) Age-related issues and loss of social support impact treatment.	Explore how older methadone patients’ experiences and views affect their use and expansion of social supports to stay abstinent.
**Vallecillo et al., 2022, Spain [43]**	Quantitative, cross-sectional design. Group comparison: MMT cohort vs. general population. N = 99; 72.7% men. Mean age: 55.7 (SD = 4.7; 50+).	Cardiovascular risk in older patients with opioid use disorder.	1. Compare cardiovascular risk factors and global risk in OUD adults aged >50 with the general population; 2. Assess the efficacy of calibrated functions for risk.
**Vikbladh et al., 2022,** **Sweden [58]**	Quantitative, retrospective survey, cross-sectional design. N = 190; 64–72% men. Mean age: 46. On-site public healthcare group (SD = 10.6, range 25–63).	(a) Age correlated with more comorbidities and cognitive impairment; (b) More pronounced chronic and severe diseases; (c) Understanding healthcare use in older patients.	1. Assess healthcare utilization for somatic conditions in OST patients; 2. Compare healthcare use among OST patients with and without on-site public healthcare.

^1^ The literature identifies several terms that describe the management of opioid dependence through medication-assisted interventions, including MMT (Methadone Maintenance Treatment), OST (Opioid Substitution Therapy), OMT (Opioid Maintenance Treatment), OAT (Opioid Agonist Therapy), AOT (Agonist Opioid Treatment), and OTP (Opioid Treatment Programme).

**Table 5 ijerph-22-00458-t005:** Summary of themes, subthemes, and results emerging from the scoping review.

Themes	Key Results
Health	
**Bech et al., 2019,** **Norway [47]**	The average age at death was 48.9 years. *Causes of death:* 45% somatic, 42% drug-induced, and 12% violent. Lower drug-induced death risk with older age and higher comorbidity. Somatic death rates doubled for patients over 50 compared to those aged 41–50. Higher mortality rate among men than women. Methadone users had twice the mortality rate of buprenorphine users. Most deaths occurred at home.
**Bech et al., 2021,** **Norway** [48]	Older age was linked to pulmonary, cardiovascular, and renal pathology, but not liver pathology. Cardiovascular and renal risks remain higher in older age, even after adjusting for BMI and sex. A total of 91% (n = 48) had a disease in at least one organ system; 76% (n = 40) had diseases affecting two or more organ systems.
**Beynon et al., 2009,** **UK [34]**	Heroin was often used together with alcohol, cannabis, LSD, amphetamine, cocaine, crack, and mushrooms. *Common physical health issues:* circulatory, respiratory, pneumonia, diabetes, hepatitis, liver cirrhosis, overdose, vein damage; about half had hepatitis C. *Prevalent mental health issues:* depression, loneliness, anxiety, cognitive impairment, dementia; short-term memory issues linked to age and substance use, including alcohol. *Negative experiences in hospital settings:* stigma, poor pain management (especially in palliative care), and low expectations of specialist services.
**Beynon et al., 2013,** **UK [35]**	Most rated their health as poor and had low-quality, limited-quantity diets, often opting for easily prepared foods, and rarely cooking due to financial and dental issues. Minimal intake of fruit and vegetables. A total of 50% of participants used illicit drugs, including heroin, crack, and cannabis, while most consumed alcohol and used prescriptions. All smoked daily. *Reported various health issues*: infections, circulatory and respiratory problems, pain, weight issues, liver cirrhosis, and headaches. BMI was not included in care planning activities.
**Grella & Lovinger, 2012, USA [46]**	*General health status*: A total of 47.8% reported fair or poor, 4% reported excellent health. *Gender difference*: More women reported poor health than men (27.3% vs. 8.4%) and faced more health conditions including heart disease, circulatory problems, asthma, bladder issues, colitis/bowel problems, arthritis, and chronic headaches, as well as higher levels of distress. *Functional score*: Lower scores in physical role functioning, bodily pain, general health, energy and fatigue, and social function compared to the general population.
**Grischott et al., 2019,** **Switzerland [50]**	Many older participants were smokers or had pulmonary symptoms. A total of 15% had undiagnosed COPD; 30% exhibited post-bronchodilator airflow limitation, indicating COPD. COPD patients were older (mean 51.0 vs. 42.6) and had greater lifetime exposure to tobacco and cannabis. Men over 40 in OAT had a 2.4 higher prevalence of COPD vs. Swiss ever-smokers. Older participants were highly open to pharmacological COPD treatment.
**Han et al., 2022,** **USA [38]**	Older OAT patients compared to the national cohort had higher rates of health issues (psychiatric disorders, chronic lung disease, cancer) and hospitalization, and greater physical challenges: falls, mobility impairment, and chronic pain
**Lintzeris et al., 2016,** **Australia [39]**	Over half reported liver disease and head injury, with common conditions including circulation issues in the legs, respiratory and gastrointestinal problems, hypertension, cardiac conditions, seizures, cancer, diabetes, and stroke. Most experienced falls in the past year, often resulting in injury or medical attention. Older patients had significantly lower physical health scores compared to those in alcohol-related treatment, indicating poorer physical health, quality of life, and a trend toward worse mental health. Few older adults received carer support, despite one-third struggling with daily tasks. A total of 15% of older patients reported violent crime victimization, compared to 5% of younger patients (ages 14–49).
**Lofwall et al., 2005,** **USA [23]**	*Increased health issues*: Older adults reported higher rates of cardiovascular, gastrointestinal, and bone/joint issues; and greater prescription medications compared to younger participants. *Lower health-related quality of life*: Older adults scored lower on measures of physical functioning, role limitations due to physical health, and bodily pain. Their scores fell below population norms in all health-related quality of life domains, including physical and social functioning, mental health, vitality, and general health perception.
**Maruyama et al., 2013,** **Canada [40]**	Compared to the general population, older adults in OAT have higher rates of treatment for COPD and depression, and higher rates of hypertension and diabetes.
**Medved et al., 2020,** **Norway [52]**	*Chronic condition:* Nearly two-thirds had at least one chronic condition: HCV—the most common (52.9%), but rarely treated. Asthma—the next most common, with 80% receiving treatment. Others: high blood pressure, heart disease, COPD, and diabetes, with over 50% receiving treatment. Over 50% reported at least seven somatic complaints in the past six months, including reduced memory, headaches, indigestion, dizziness, dental issues, constipation, and joint pain. In this period, 81% visited a GP, and over 50% had additional somatic care visits. More than 60% were satisfied with OAT.
**Ramli et al., 2019,** **Malaysia [55]**	A total of 54% attempted ED treatment, 80% found it effective. A total of 48% did not seek information; 42% thought that “ED is not a serious condition”. No significant differences in effectiveness were noted across medical, self, and alternative treatments.
**Ramli et al., 2020,** **Malaysia [56]**	A total of 47.6% were sexually inactive, linked to older age and being single or divorced; no association with comorbidities, medications, or dose and duration of methadone treatment. Participants had hepatitis C and B, hypertension, HIV, and diabetes mellitus; 1.5% received hepatitis medication.
**Rosen et al., 2008,** **USA [41]**	*Mental health*: Depression is the most common disorder; post-traumatic stress disorder and generalized anxiety disorder are the most prevalent anxiety disorders. A total of 47% were on psychotropic medication for mental health issues. Women were twice as likely to have agoraphobia compared to men (20.8% vs. 9.8%). *Physical health*: 58% reported fair to poor physical health. Common health issues included arthritis (54.3%), hypertension (45%, significantly higher in men), and hepatitis C (49%). *Age and health outcomes*: OAT adults aged 50–54 had worse health outcomes than those aged 55–67 in the general population. *Abstinence*: Positive screening rates for substances remained similar one year after the study to pre-study levels, with 61.4% of respondents recording at least one positive result.
**Vallecillo et al., 2022, Spain [43]**	Health risks in older adults with OUD compared to the general population: higher prevalence of cardiovascular risk and increased incidence of tobacco smoking. In addition, common low HDL cholesterol, hypertriglyceridemia, atherogenic dyslipidaemia, and a high prevalence of abdominal obesity.
**Treatment**	
**Ayres et al., 2012,** **UK [32]**	*Relationships with professionals*: Positive relationships with healthcare professionals. Satisfied with prescriptions; preferred weekly collection over daily. Fear of losing prescriptions with GP changes, and supervised intake is viewed as humiliating. *Treatment challenges*: Poor hospital/dental care experiences, and unmet health needs. Limited access to additional treatment. *Age-related concerns*: Shame about drug use due to age, and age as a barrier to seeking help. Detoxification challenges with age and a need for age-specific services for older, stable OAT patients.
**Choi et al., 2022,** **USA [36]**	*Patterns by group*: Racial/ethnic minorities had lower odds of completing outpatient treatment. The heroin group had a higher rate of cocaine/crack use; more often included in treatment plans; legal referrals predicted treatment completion. The prescribed opioids group had higher rates of benzodiazepine/tranquillizer use and having a bachelor’s degree increased the odds of completing treatment. *Influence of OAT*: OAT was linked to higher odds of completing residential treatment but lower odds of completing detoxification and outpatient treatment. *Outcomes by age*: Older adults (65+) with prescribed opioids had higher odds of completing residential treatment than those aged 55–64.
**Fareed et al., 2009,** **USA [49]**	Retention in methadone treatment reduced drug-related, psychiatric, medical, and legal issues, with a trend toward fewer alcohol-related problems. Discontinuation of treatment did not demonstrate improvements, except for a slight reduction in family issues. Targeting interventions aimed at lifestyle risk factors and comorbid medical conditions, such as nicotine dependence and diabetes mellitus, could enhance health outcomes for older adults with opioid dependence. Premature death linked to diabetes and cancer before age 65.
**Han et al., 2024,** **USA [44]**	Chronic diseases were common, with 58% having hypertension, 25% hepatitis C, and 89% experiencing multiple chronic diseases. As they aged, participants often sought healthcare outside their OAT but faced discrimination, leading to mistrust of the system, hesitation to disclose substance use, and delays in routine care or reliance on emergency departments. Participants faced challenges ageing with OUD, including health declines, housing insecurity, and limited access to care due to transportation barriers. Reflecting on mortality sometimes motivated well-being efforts. Many favoured OATs for integrated services due to familiarity and reduced stigma.
**Vikbladh et al., 2022,** **Sweden [58]**	Primary healthcare utilization was associated with older age and being born in Sweden. Greater physical health concerns among on-site vs. regular primary healthcare users. *Physical concerns*: 52% had musculoskeletal diagnoses, one-third of the on-site group had gastrointestinal diagnoses, including constipation, and 62% reported neck, back, or extreme pain. In contact with secondary care: 84% of on-site users vs. 64% of regular users.
**Demographics**	
**Badrakalimuthu** **et al., 2012,** **UK [33]**	A total of 10% of newly enrolled patients were aged ≥60; the average first service contact at 41.4 years (range 21–65). A total of 11% initiated opiate use at age ≥ 50; 24% first contacted services at this age. *Substance use*: 73% methadone, 21% buprenorphine; 60% other substances; 31% multiple substances (82% used only two, mainly cannabis, benzodiazepines, or cocaine). A total of 66% had other prescriptions. *High comorbidity*: physical (54%) and mental (62%). A total of 37% had HCV. Treatment compliance was 86%. *Participants under 50 years* vs. *aged ≥50*: often single; unstable housing; higher rates of buprenorphine use; high-dose prescriptions; lower rates of blood-borne viruses and health issues.
**Han et al., 2015,** **USA [51]**	*Demographic shift*: Adults ≥50 constituted the majority in treatment over time, with an increasing proportion of females aged 50–59. Among patients over 60: high prevalence of White individuals compared to Black individuals. Higher rates of sight and mobility impairments among adults aged ≥50 vs. younger groups.
**Pierce et al., 2018,** **UK [54]**	The methadone-specific DRD rate was seven per 10,000 person-years overall, varying by age: 3.5 for 18–34, 8.9 for 35–44, 18 for 45+. The heroin-specific DRD rate was 12.5 per 10,000 person-years, higher in males. Being out of treatment doubled the risk of heroin-specific DRD and nearly quadrupled it, with a non-significant 20% decrease in methadone-specific DRD risk. Hazard ratios for methadone-specific deaths compared to ages 25–34 were 0.87 for under 25, 2.14 for 35–44, and 3.75 for 45+.
**Social aspects**	
**Conner et al., 2008,** **USA [37]**	*Participants faced stigma in eight categories*: drug addiction, ageing, psychotropic medications, depression, OAT, poverty, race, and status. Stigma with OAT, injection drug use, and age were most common. Men reported more stigmas than women. Stigma came from family, friends, rehab staff, and counsellors. *Most participants faced multiple types of stigmas*: 33% drug addiction and ageing; 25% depression and psychotropic medications; 33% drug addiction, OAT, and psychotropic medications; or drug addiction, ageing, and poverty; 66% drug addiction, ageing, psychotropic medications, and depression. Stigma and fear of treatment, including from psychiatrists, hindered service use. Age and poverty restricted access to mental health care.
**Shen et al., 2018,** **China [57]**	*Composition of networks*: Over 50% had parents in their social network, while one-third had friends or spouses. A total of 41% had older adults in their networks; 32% had individuals of the opposite sex. A total of 92% had at least one very close network member, most of whom provided financial or emotional support. A total of 24% had at least one network member who used substances with them. The majority reported positive or mixed experiences with clinic counsellors.
**Smith & Rosen, 2009,** **USA [42]**	Reluctance to trust others hindered social support expansion among older methadone clients. *Key barriers*: Trust issues: personal feelings and guilt impeded trust and relationship formation with staff, peers, and others. Personal loss: death of family and friends due to illness, violence, or drug-related causes. Relationship strain: conflict and abuse, particularly with family members using illegal drugs and turbulent intimate relationships. Self-isolating common among participants; limited social support.
**Substance use**	
**Grella & Lovinger, 2011, USA [45]**	*Rapid decrease* (25%): Highest female proportion (60%), least school-related issues (27%), older onset, lowest cocaine use (6%) but highest amphetamine (10%) and alcohol use (40%). *Moderate decrease* (15%): Fewest with conduct disorder (32%); decreased odds of heroin use *Gradual decrease* (35%): Highest rates of conduct disorder (52%) and antisocial personality disorder (49%), most time incarcerated (20%), youngest age of onset. *No decrease* (25%): Longest duration in OAT (38%). Low drug use (18%), late-onset increase (23%), early-onset increase (25%), gradual decrease (18%), and no decrease (16%). Heroin use varied among drug use groups, indicating complex patterns of heroin and other drug use. No single dominant pattern across the sample.
**Nyamathi et al., 2009,** **USA [53]**	*Substance use*: 51% heavy alcohol use, 46% heroin. In the past 30 days, older adults reported use of cocaine, marijuana, barbiturates, amphetamines, and hallucinogens. *Heavy drinkers*: had the highest rates of victimization; over half reported poor physical and emotional health; about half engaged in heavy drug use, unlike moderate drinkers. Older adults in fair or poor health were more than three times more likely to drink heavily than those in good to excellent health.

## Data Availability

The raw data supporting the conclusions of this article will be made available by the authors on request.

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
