# Peer review of "Individuals 45 Years and Older in Opioid Agonist Treatment: A Scoping Review"

_ijerph, 2025, doi:10.3390/ijerph22030458_

Round 1
Reviewer 1 Report
Comments and Suggestions for Authors
Ageing individuals in opioid agonist treatment: a scoping review (ijerph-3356885)
The manuscript describes results from a scoping review of articles that have examined opioid agonist treatment (OAT) in older adults (ages 45+, on average). A total of 28 empirical studies met inclusion criteria. Results spanned a variety of areas that served as the foci of the original studies, including health effects, barriers to treatment, stigma, use of other substances including alcohol and benzodiazepines, and relationships between demographic factors and disease and treatment outcomes. Overall, the results are interesting and the article would be a good resource for researchers and clinicians. Strengths include the systematic approach to study identification, analysis of an interesting set of studies and strong writing. Limitations include some aspects of the impact of inclusion criteria on study findings.
Comments below may serve to strengthen the manuscript.
1. Many of the reviewed studies exclusively recruited older adults (e.g., 50 years or older) whereas some other studies were included that recruited adults across the lifespan, including much younger adults – these studies were retained in the analysis if the average age of participants was at least 45 years of age. It was unclear in the review of the latter studies if the results applied exclusively to the subset of participants who were over 45 years of age or if the results summarized applied to all participants in those studies, regardless of age?
If the summary includes all participants, to what extent is that an accurate reflection of older adults in OAT treatment?
2. A minor point, but the final sentence should probably say “rise” rather than “rice.”
Reviewer 2 Report
Comments and Suggestions for Authors
RE: Aging individuals in opioid agonist treatment: a scoping review
This is a scoping review examining individuals aged 45 and above undergoing opioid agonist therapy (OAT). The review aims to identify the characteristics of middle-aged and older adults in OAT, explore relevant social, psychological, and medical factors among this population, identify gaps in research, and offer recommendations. The inclusion criteria focus on studies involving middle-aged and older adults (aged 45 and above) undergoing treatment for opioid use disorders (OUD) in inpatient or outpatient settings. Eligible studies include observational and qualitative research published in English, Norwegian, Swedish, Danish, or German, with no restrictions on publication timeframes. Additionally, abstracts, posters, editorials, reviews, reports, grey literature, meta-analyses, case studies, and interventional studies are excluded, as are studies not directly related to aging within the OAT context.
The main strength of this scoping review lies in its contribution to an area of research that remains significantly understudied, and becomes relevant as people with OUD are aging while undergoing OAT. There is an historical lack of attention to this populations and a scarcity of outcome research targeting this demographic as well as not sufficiently recognized in policy or practice. This scoping review provides a synthesis of evidence related to the characteristics, challenges, and care needs of older adults with OUD.
The review provides a valuable effort to consolidate information on an understudied topic; however, several methodological concerns need to be addressed to strengthen its impact. One key limitation is the lack of a guiding theoretical framework, which would provide a structured lens for analyzing and interpreting the findings. Without this foundation, the synthesis presented in the review risks feeling like a collection of categorized information rather than a meaningful integration of insights that advance understanding.
The discussion, as it stands, tends to organize existing knowledge into familiar categories without offering new or transformative perspectives. This approach does little to deepen our understanding of the nuanced interplay between opioid use and aging. For example, the review could benefit from a stronger exploration of intersectionality—examining how aging interacts with social, psychological, and medical dimensions unique to older adults in OAT. By digging deeper into these intersections, the review would move beyond simply restating known categories and instead provide fresh insights into the complex realities of this population. Incorporating a more analytical and intersectional approach would elevate the review, ensuring it goes beyond organizing information to truly synthesizing it in a way that informs future research, policy, and practice.
Specific comments:
Title:
Does it reflect the inclusion and exclusion criteria?
Abstract:
If changes are made to the manuscript, the abstract can be improved to reflect them.
Introduction:
What is the context of the review? Who is the audience of this paper?
Methods
The two sentences provided define the population differently:
1. "Studies were included if they focused on age and age-related outcomes in their objectives or findings, with participants having a mean age of 45 or older. (page 3, line 73-75)"
This criterion allows for the inclusion of studies where younger participants are present, as long as the overall mean age meets the threshold of 45 or older. This could result in studies with a significant portion of younger participants being included, potentially diluting the focus on middle-aged and older adults.
2. "Middle-aged and older adults (aged 45 and above) undergoing treatment for OUD. Exclusion: Individuals under the age of 45 (Table 1)"
This sentence explicitly excludes any studies with participants younger than 45, ensuring that only studies exclusively focused on the 45+ population are considered. This stricter criterion conflicts with the first sentence, which allows younger participants as long as the mean age meets the threshold.
This inconsistency needs to be resolved and clarified
The search in table 2 might not fully reflect the terminology used in North America, where opioid agonist treatment is more common, as well as opioid use disorder, diacetylmorphine, morphine, hydromorphone, suboxone, etc.
Some exclusions are presented to enhance rigor and are well-justified, but others lack sufficient reasoning and may unnecessarily limit the scope of the review. For example, the exclusion of reviews, justified by avoiding duplication and focusing on primary research, is problematic. Reviews synthesize existing evidence and identify research gaps, which are essential for scoping reviews aiming to contextualize findings. Including high-quality reviews can strengthen the analysis rather than undermine it. Similarly, the decision to include only studies in English, German, and Scandinavian languages due to the research team’s language proficiency overlooks the availability of translation tools, which can mitigate language barriers. Translation inaccuracies are generally outweighed by the benefit of accessing potentially critical evidence from diverse linguistic sources. Excluding non-translated works may unintentionally narrow the evidence base, particularly in a globalized research landscape where significant findings are often published in languages outside these parameters.
“Limited attention has been given to gender differences” Are the studies looking at health, not accounting for sex and/or gender?
Discussion
In the discussion, the authors say that the study “examine the characteristics and factors influencing older adults”. What does this mean?
The discussion is written in a way that at times the topic of aging is not present or evident at all. There is no intersectionality and well-known issues associated with OUD are carried forward, without a thoughtful integration.
Recommendations are really high level. The authors indicated they have a team with diverse expertise. Thinking about who is the audience of this paper, what actionable recommendations would you suggest?
Reviewer 3 Report
Comments and Suggestions for Authors
Thank you for the opportunity to review this timely and interesting scoping review. The subject matter is certainly of growing concern, especially for nations with high rates of opioid use disorder. Data collection table showed that this is a subject that is receiving more attention, and your analysis points out some poignant gaps in current research that warrant further attention. I have some questions and comments for some minor revisions and refinements that I believe will clarify the flow of the information.
1. Introduction: The introduction is concise and clear. There are comments in the Discussion that could trigger some editing in the introduction section for the sake of greater thematic continuity.
2. Introduction, Page 1, lines 25-26: You list the average age increase for OAT patients in Norway, but most of the studies are from the US. It would help the justification to show that this is a larger trend.
3. Materials and Methods, general: This section is well-organized and nicely written.
4. Material and Methods, Page 4, Figure 1: I’m not clear on what “not retrieved” means in this context.
5. Results, general: Most of the sub-sections of the Results could be more heavily referenced (i.e., when you write about a series of topics addressed across many papers, reference which paper mentioned each topic). It would also be good to be more specific about how many papers addressed which subject in the results narrative. For instance, on page five, in the following paragraph: Even though research aims differed, several studies [how many and which ones?] focused on examining age-related health outcomes, such as mortality rates and chronic conditions. Others [how many and which ones?] assessed healthcare utilization and proposed considered potential modifications to current OAT practices to better serve an ageing population.
This is just one example. Please read and edit the results narrative with this in mind and add detail and citations accordingly.
6. Results, Page 12, line 202: Do you mean “abstinent” here, or “absent” as in they dropped out of the study?
7. Results, Table 5: I am having some issues interpreting this table. Does each row correspond to just one of the included studies, or are the key results synthesized between studies according to the theme?
â—¦ If each row is showing key results of just one study, then perhaps you could just list the author and year in the same cell as the sub-theme name.
â—¦ Not being sure that you were utilizing just one study per row in this table, the repetition of sub-themes was confusing. For instance, “somatics” is listed twice as a sub-theme, with different key elements, apparently from different papers, but the two sub-themes are not grouped together, so its a little confusing.
â—¦ In some ways, it would be easier to read the table if you just listed the sub-themes once and combined all relevant results from each study into the key elements cell, citing each specific element to the study from which it came.
â—¦ However the authors determine to organize this very long and important table, it would be helpful if you were to describe your method of constructing that table so readers could more easily interpret what they are seeing.
â—¦ I also question how the authors are defining “somatic” and why it is listed separately from other types of physical health issues in the sub-themes.
8. Discussion, general: The discussion section could use a little more polishing and detail. There are many missing references, and some meandering subject matter that could be better focused for better effect. Specifically, I would suggest being more decisive about when you are discussing the results of this review as opposed to discussing what was NOT present in the literature. If the journal formatting allows, you might consider making sub-topics within the discussion arranged around your original aims for the paper, as numerated at the end of the introduction.
9. Discussion, page 16, line 209: Would it be more appropriate to say that you found five themes emergent from the included studies? You didn’t determine the themes ahead of time, correct?
10. Discussion, page 16, lines 227-236: This paragraph needs citations.
11. Discussion, page 16, lines 244-256: These paragraphs needs further citation. Also, the one reference at the end of the first paragraph is not in the same format as the rest of the paper.
12. Discussion, page 17, lines 269-281: This paragraph opens with an exhortation to more closely examine psycho-social dimensions of life for older people using OAT, then branches off into a discussion of how the various health systems vary across nations and the impact of that variance on patients’ economic stability and mental health. It’s a little scattered.
13. Discussion, general: It is not always clear when you are discussing the results of this review and when you are branching beyond the included studies into other literature.
14. Discussion, general: You have developed the topic of economic issues facing older populations using OAT, yet there was no mention of economic issues in the introduction, and none of the included papers seem to touch on the subject. If you are going to bring it into discussion, please consider including it in the Introductory section as a subject you are interested in exploring that may not have been broached in the literature.
15. Conclusion, page 19, line 371: I believe you mean “rise” instead of “rice”.
After making the revisions, the authors should carefully review the manuscript for any grammatical and stylistic issues.
